# Synchronizing Verbal Responses and Board Writing for Multimodal Math Instruction with LLMs

**Yuan-Hao Jiang**[1,2,4,5], **Ruijia Li**[6,7], **Yuang Wei**[1,2,3,5], **Rui Jia**[1,2,5],
**Xiaobao Shao**[1], **Hanglei Hu**[1], **Bo Jiang**[1,2,*]

[1] School of Computer Science and Technology, East China Normal University
[2] Lab of Artificial Intelligence for Education, East China Normal University
[3] School of Computing, National University of Singapore
[4] Graduate School, Shanghai Jiao Tong University
[5] Shanghai Institute of Artificial Intelligence for Education, East China Normal University
[6] Faculty of Education, East China Normal University
[7] Institute of Artificial Intelligence, China Telecom

## Abstract

The advancement of large language models (LLMs) has greatly facilitated math instruction, with the generated textual content serving as verbal responses to address student inquiries. However, in instructional settings, teachers often provide both verbal responses and board writing (BW) simultaneously to enhance students' knowledge construction. To address this, we introduce MathBoard, a multimodal large language model (MLLM) designed for elementary mathematics education, which progressively generates BW. Our study focuses on the provision of BW to learners, aiming to reduce their cognitive load effectively. Furthermore, MathBoard can be integrated with other approaches that enhance mathematical reasoning capabilities. An empirical study involving 34 pre-service teachers demonstrated that the multimodal interactions facilitated by MathBoard were more highly accepted and impactful across various dimensions compared to text-only interactions, significantly promoting learners' social construction of knowledge.

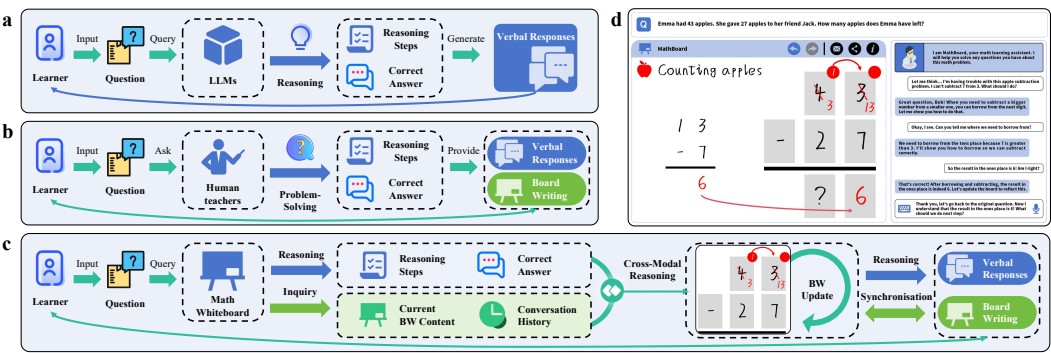

Figure 1: The cross-modal reasoning process of MathBoard in solving mathematical problems and its user interface design. In (a), (b), and (c), the reasoning details of LLMs, human teachers, and MathBoard in assisting students with solving mathematical problems are presented, respectively. (d) also illustrates the user interface of MathBoard.

*Corresponding Author: bjiang@deit.ecnu.edu.cn

38th Conference on Neural Information Processing Systems (NeurIPS 2024).

# 1 Introduction

In recent years, LLMs have shown immense potential in natural language processing and have played significant roles across multiple disciplines, particularly in mathematics(1; 2). LLMs can automatically generate exercises, provide instructional support, and deliver personalized feedback for students(3; 4). For specific educational needs, models like EduChat offer personalized, equitable, and empathetic services through fine-tuning(5), while LoRA fine-tuning strategies facilitate the automation of educational data annotation(6). The MinT model focuses on enhancing logical reasoning and generalization abilities(7). Looking ahead, further efforts to improve the sustainability and interpretability of LLMs will be essential for enhancing their trustworthiness and reliability in educational contexts(8; 9).

The concept of shared whiteboards has proven effective in improving efficiency in collaborative teams(10). Recent research has focused on integrating LLMs into whiteboard collaboration environments to promote creative cooperation and problem-solving through Blackboard Writing (BW) technology. For instance, the AI-AB framework provides an interactive whiteboard platform that facilitates idea exchange between humans and LLMs(11). Related works include the Visual Sketchpad, which allows LLMs to add auxiliary lines when solving mathematical problems(12), and the Whiteboard-of-Thought project, which demonstrates how LLMs can improve their OCR performance on whiteboards by enhancing reasoning abilities(13). While these studies primarily focus on generating Python code to improve LLMs' reasoning capabilities, they do not directly serve as visual teaching aids for human learners. Therefore, we propose further exploration into the generative capabilities of LLMs to create cross-modal learning resources, potentially transforming human-computer interaction models and providing learners with a more personalized and intuitive learning experience.

The primary contribution of this study is the development of MathBoard, powered by LLMs, which synchronously generates both Verbal Responses and Board Writing, thereby offering learners a cross-modal mathematics learning experience. However, a current limitation of MathBoard is its applicability solely to elementary-level math instruction, which requires further refinement in future work. This study seeks to address the following research questions:

- How can the generative capabilities of LLMs be leveraged to provide cross-modal guidance in mathematics learning?

- Is the proposed cross-modal teaching method more acceptable and engaging for learners?

- Does the integration of Board Writing in mathematics instruction foster learners' social construction of knowledge?

# 2 Related Work

## 2.1 Multimodal Large Language Models for Education

LLMs have seen widespread application in education (14; 15; 16). With the rapid advancements in Multimodal Large Language Models (MLLMs), numerous educational case studies have highlighted their effectiveness and potential utility (17; 18; 19). For instance, MLLMs are capable of generating multimodal writing suggestions through diverse channels, including text, visuals, and audio, thereby aiding learners in enhancing their writing proficiency (20). Additionally, MLLMs can integrate multimodal data collected during classroom activities to produce more precise transcriptions, facilitating post-class study or reference (21), as well as to assess student engagement and evaluate the effectiveness of educational resources and environments (22). MLLMs also have the potential to provide interpretable information in education(23; 24). Notably, given MLLMs' advanced capabilities in processing multimedia information, they hold significant promise for supporting visually impaired learners in acquiring knowledge and understanding the world around them (25). Although the deployment of MLLMs necessitates increased data exchange(26; 27), which may pose potential security risks, techniques such as Federated Learning offer a viable means to mitigate these concerns (25; 28; 29).

## 2.2 MLLMs for Math Learning

In mathematical problem-solving, reasoning skills are essential(30; 31; 32). Additionally, given that mathematical problems often include charts and data, the ability to process multimodal information is also necessary(33; 34; 35). While the integration of multimodal data inputs can provide MLLMs with richer information and greater problem-solving potential, research has demonstrated that many MLLMs struggle to accurately interpret charts within the problem-solving context, leading to the ineffective utilization of multimodal information (36).

To improve MLLMs' comprehension of such data, one effective approach is the use of text-based question-answer pairs to redraw geometric figures, thereby enhancing their understanding of geometric problems (37). This approach essentially transforms multimodal data into pure textual information, making it more accessible for MLLMs. Moreover, several other strategies have been employed to boost MLLMs' problem-solving capabilities: the introduction of skill example repositories (38), fine-tuning models using chart data embedded in mathematical problems (39). Additionally, designing reasoning path retrieval methods suitable for multimodal mathematical problems is crucial for MLLMs. These methods include tree-based multimodal reasoning path searches (40) and guided extraction of key information tailored for solving lengthy mathematical problems (41). These methods have improved MLLMs' understanding and problem-solving abilities in mathematics, but we believe it is even more crucial to integrate these reasoning results effectively into mathematics education and tutoring(42). Therefore, an approach that complements these studies is still needed to provide learners with a multimodal learning experience.

# 3 MathBoard

Many existing studies focus on enhancing the reasoning capabilities of LLMs in solving mathematical problems. However, our research emphasizes improving the learning experience and reducing cognitive load by utilizing a visualized BW. To this end, we developed the MathBoard. In real-world classrooms, mathematics instructors frequently provide verbal explanations in tandem with BW illustrations to guide students through problem-solving processes. For example, a teacher might say, "Notice that we need to borrow from the tens place to the ones place. This changes the tens digit from 4 to 3, and the ones digit from 3 to 13, like this." Simultaneously, the teacher would draw an arrow on the whiteboard from the tens to the ones place, alter the 4 in the tens place to 3, and update the 3 in the ones place to 13. While current methodologies predominantly focus on enhancing LLMs' reasoning capabilities, they lack mechanisms for progressive BW generation. We illustrate this process in Figure 1, where Figures 1(a), 1(b), and 1(c) present the detailed reasoning pathways employed by LLMs, human teachers, and the proposed MathBoard, respectively, during mathematical problem-solving.

In detail, MathBoard first generates the reasoning process for a given mathematical problem, producing both the reasoning steps and the correct solution while simultaneously querying the current BW content and conversation history. These components are then used for cross-modal reasoning. If responding to the problem for the first time, the system creates a new BW; otherwise, it updates the existing BW, enabling a progressive generation of the visual content. This iterative process results in a synchronized update of both the BW and the verbal response, which together help learners independently resolve the mathematical problem. It is important to clarify that the proposed method is orthogonal to existing approaches aimed at enhancing the reasoning capabilities of LLMs. These existing methods can be effectively applied during the initial reasoning process conducted by Math-Board, facilitating the attainment of more accurate answers and a more detailed reasoning steps. Subsequently, MathBoard can integrate these components for the ensuing cross-modal BW reasoning.

Additionally, Figure 1(d) presents the user interface of the MathBoard, comprising three main sections: the Board Writing Area, Chat Area, and Input Question Area. Learners input mathematical problems in the Input Question Area and interact with the MathBoard in the Chat Area. With each system response, both the verbal response and the updated BW are synchronously provided, with the verbal response displayed in the Chat Area and the BW update rendered in the Board Writing Area. Learners can continue interacting with MathBoard via the Chat Area until the problem is fully resolved. Detailed information regarding the case study of MathBoard can be found in Appendix A.

## 4    Design of experiments

The study used the Educational Technology Acceptance & Satisfaction Model (ETAS-M)(43) to design a questionnaire, assessing system performance, including improvements in learning efficiency, speed of task completion, and ease of understanding complex concepts. The study also discussed the accuracy of the information provided by the system, the design of the operation interface, and stability, as well as the role of the system in promoting student interaction, group activity participation, and improvement in understanding. Further details regarding the experiment can be found in Appendix B.

## 5    Results

### 5.1    Reliability and Validity Analysis

The reliability analysis of the subjects' scale data yielded a Cronbach's alpha of 0.947 for the entire scale, which consists of 30 questions, indicating good internal consistency and suggesting that the subjects' understanding of the scale was consistent. The Cronbach's alpha values decreased after the deletion of all question items except for the dialog rounds interaction data, indicating that no questions needed to be eliminated. Furthermore, to analyze the overall validity of the scale, the agreement between each item and the total was assessed using Pearson's correlation coefficient, all of which were positively correlated. Generally, a correlation coefficient greater than 0.6 is considered high, greater than 0.4 is moderate, and greater than 0.2 is low. In this set of 30 questions, a total of 20 items showed high correlation, and 6 items showed moderate correlation, indicating that the scale has high internal consistency and both analytical reliability and validity.

### 5.2    Evaluation of MathBoard

To investigate the actual pedagogical effectiveness and subject acceptance of the scheme proposed in this study, data from two groups of experimental subjects on ten dimensions were cross-analyzed. Group A is the control group, which uses only text interaction for math learning, and Group B is the experimental group that uses cross-modal MathBoard learning. The analysis results show that Group B scored higher than Group A on all dimensions, indicating that the visual presentation and communication approach enhances students' willingness to participate in group activities and construct knowledge in authentic contexts. To further explore the effectiveness of the program, independent samples t-tests were conducted on the dimensions of the experimental and control groups. The results show that the social constructivism dimension reached statistical significance (p=0.010), indicating that the system can significantly promote students' willingness to communicate and can be used as an auxiliary tool for students' group activities and team discussions during their studies.

### 5.3    Acceptance Variability Analysis

To further investigate whether the acceptance of the cross-modal interactive tutoring scheme proposed in this study varies among groups with different characteristics, data on the teaching experience and gender of the subjects were collected. This was done to explore and analyze whether these variables influence the teaching effectiveness of the platform. Regarding the gender variable, independent samples t-tests were conducted on the scores of male and female subject groups across different dimensions. The results indicate that there are no significant differences between the two gender groups on any dimension, suggesting that the platform's effectiveness is consistent across different genders, with no gender bias present. Additionally, for teaching experience, a one-way ANOVA was conducted with teaching experience (1-7) as the independent variable. It was found that there are no significant differences across different teaching experience groups on any dimension. This suggests that both experienced and less experienced groups show no significant difference in acceptance of the platform. In summary, it can be concluded that the platform does not produce biased effects on different subject groups.

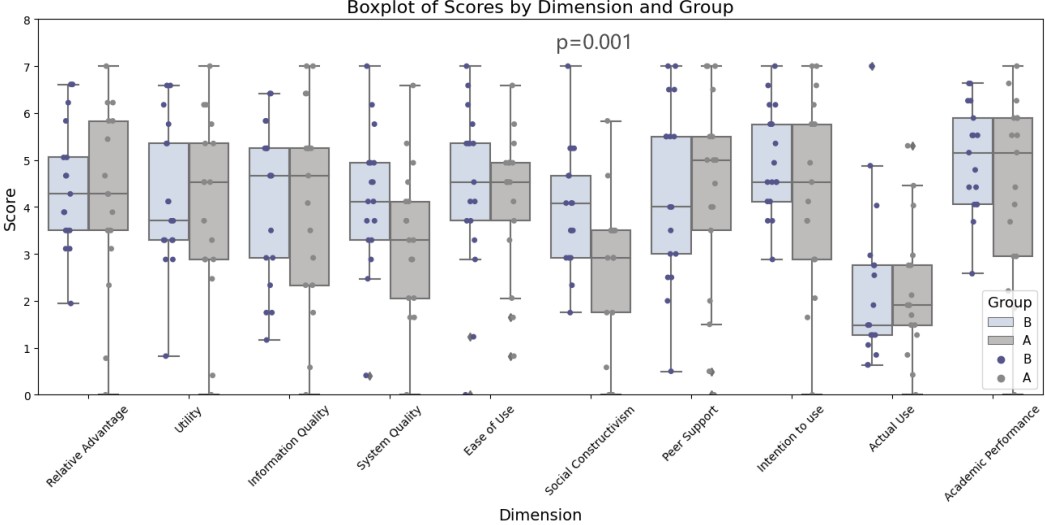

Figure 2: Comparison of platform acceptance

# 6 Discussion and conclusion

This study, through comparative analysis, found that Group B, which used cross-modal learning tools, performed better than Group A across various learning dimensions. This result supports the cross-modal Learning Theory, which posits that the combination of visual and textual elements can enhance learners' information processing and memory retention capabilities. Additionally, the Social Constructivism theory also explains Group B's superior performance, emphasizing the role of social interaction and cultural tools in knowledge construction. The study also pointed out that although cross-modal learning tools have significant advantages in promoting communication and collaboration, their effects may not be as pronounced in other areas, such as information quality or system quality.

The study offers recommendations for educational practice, highlighting the importance of integrating cross-modal learning tools into instructional design to enhance student engagement and motivation. It also suggests that educational policymakers consider investing in cross-modal learning technology when allocating resources and support the promotion of these tools through teacher training and curriculum development. These tools can not only supplement traditional teaching methods but also provide students with a richer learning experience.

Although the study's results are enlightening, there are some limitations, such as the small sample size that may affect the generalizability of the findings, and the study mainly focused on short-term learning outcomes. Future research should expand the sample size, explore the long-term effects of cross-modal learning tools in different subjects and educational environments, and how to promote the effective integration and application of these tools through educational policies and teacher professional development. Through these efforts, a better understanding of the potential of cross-modal learning tools can be achieved, and they can be utilized to enhance educational quality and the learning experience.

# 7 Acknowledgments

This work was partially supported by the National Natural Science Foundation of China, under Grant 62477012, and the Natural Science Foundation of Shanghai, under Grant 23ZR1418500, and the Special Foundation for Interdisciplinary Talent Training in "AI Empowered Psychology / Education" of the School of Computer Science and Technology, East China Normal University, under the Grant 2024JCRC-03, and the Doctoral Research and Innovation Foundation of the School of Computer Science and Technology, East China Normal University, under the Grant 2023KYCX-03.

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

## A    Case Study

The following content presents the details of MathBoard in the case of subtraction, illustrating a complete dialogue workflow. Due to space limitations, we only showcase the content of MathBoard's Board Writing (BW) for the key steps. In practice, each verbal response is generated synchronously with the BW. Figures 3 and 4 display the first and second halves of this case, respectively.

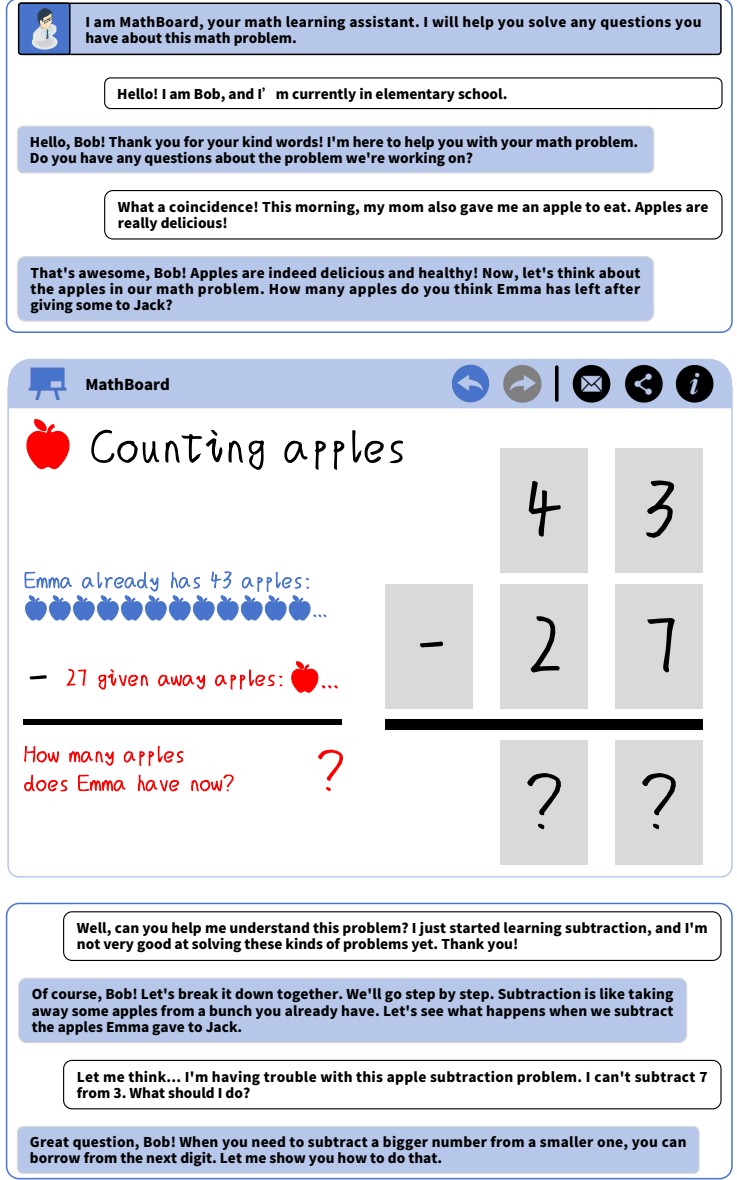

Figure 3: Details of the first half of MathBoard in a subtraction case.

## B    Experimental Setup

This study comprehensively evaluates the MathBoard teaching aid based on the Educational Technology Acceptance & Satisfaction Model (ETAS-M)(43). The research process is designed rigorously, initially recruiting 60 pre-service teachers from normal colleges through recruitment notices. Participants have diverse backgrounds, including undergraduate, master's, and doctoral students, covering various stages of education. The Richter Scale was used to assess the participants' teaching experi-

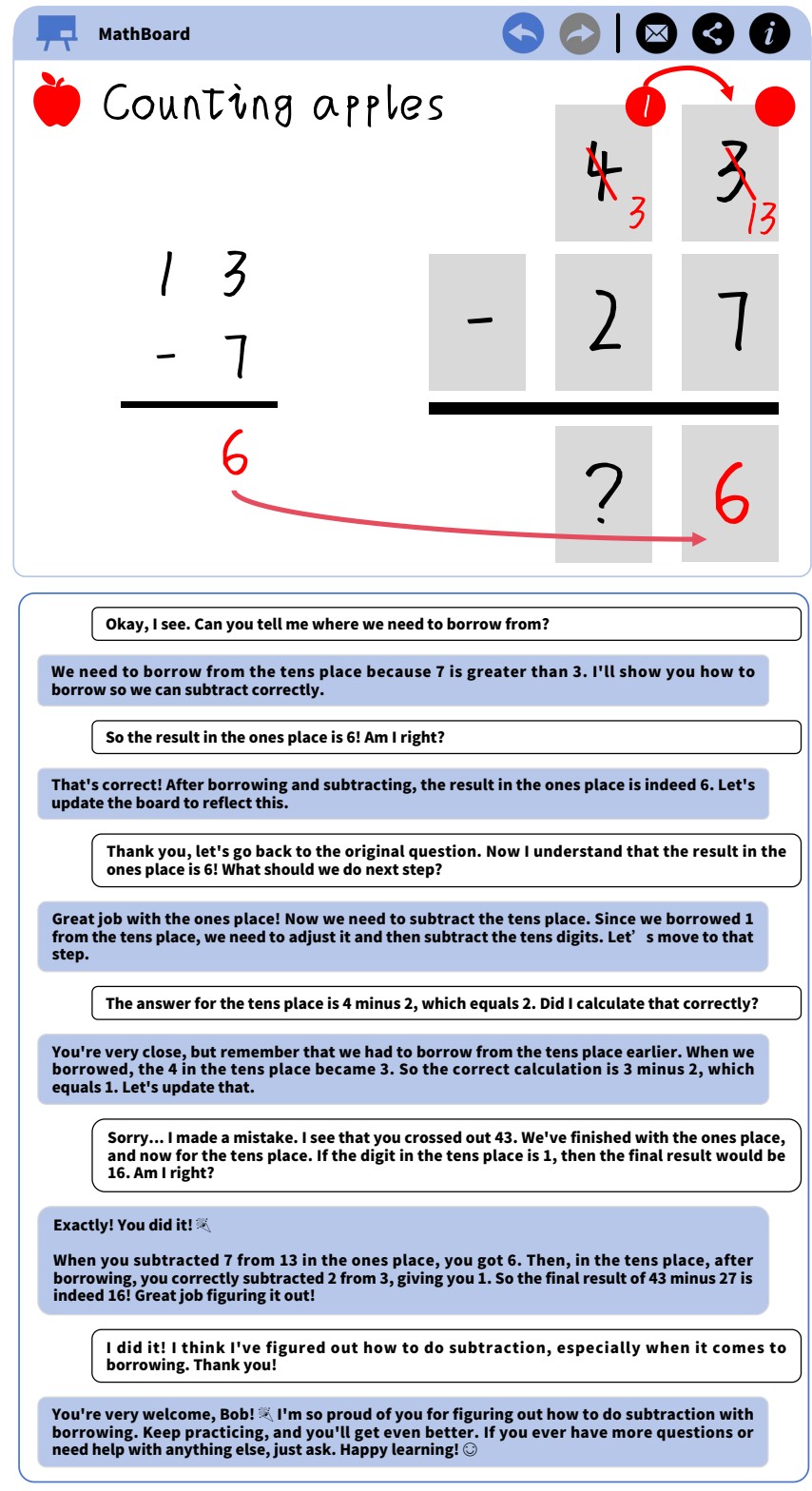

Okay, I see. Can you tell me where we need to borrow from?

We need to borrow from the tens place because 7 is greater than 3. I'll show you how to borrow so we can subtract correctly.

So the result in the ones place is 6! Am I right?

That's correct! After borrowing and subtracting, the result in the ones place is indeed 6. Let's update the board to reflect this.

Thank you, let's go back to the original question. Now I understand that the result in the ones place is 6! What should we do next step?

Great job with the ones place! Now we need to subtract the tens place. Since we borrowed 1 from the tens place, we need to adjust it and then subtract the tens digits. Let's move to that step.

The answer for the tens place is 4 minus 2, which equals 2. Did I calculate that correctly?

You're very close, but remember that we had to borrow from the tens place earlier. When we borrowed, the 4 in the tens place became 3. So the correct calculation is 3 minus 2, which equals 1. Let's update that.

Sorry... I made a mistake. I see that you crossed out 43. We've finished with the ones place, and now for the tens place. If the digit in the tens place is 1, then the final result would be 16. Am I right?

Exactly! You did it! ✏️

When you subtracted 7 from 13 in the ones place, you got 6. Then, in the tens place, after borrowing, you correctly subtracted 2 from 3, giving you 1. So the final result of 43 minus 27 is indeed 16! Great job figuring it out!

I did it! I think I've figured out how to do subtraction, especially when it comes to borrowing. Thank you!

You're very welcome, Bob! ✏️ I'm so proud of you for figuring out how to do subtraction with borrowing. Keep practicing, and you'll get even better. If you ever have more questions or need help with anything else, just ask. Happy learning! ☺

Figure 4: Details of the second half of MathBoard in a subtraction case.

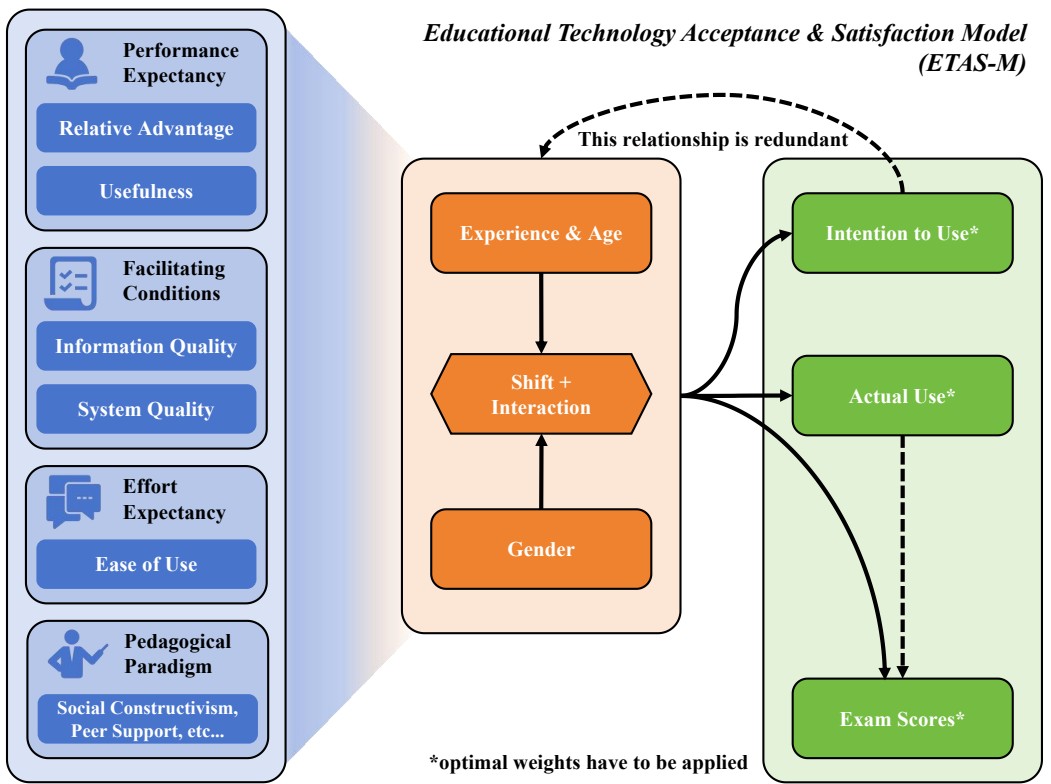

Figure 5: The ETAS-M, designed based on the UTAUT model, was created by Wessa P.(43). It takes into account the influences from performance expectancy, facilitating conditions, effort expectancy, and the pedagogical paradigm, and posits that these factors affect intention to use and actual use, ultimately impacting exam scores.

ence, with 25% of the participants having extensive teaching experience. All participants voluntarily joined the study and signed informed consent forms. The research adheres to ethical standards and has been approved by the ethics committee. The details of ETAS-M are provided in Figure 5. It is important to note that, although ETAS-M identifies gender as a potential factor influencing outcomes, our experiments did not reveal any significant differences between genders. To gather gender information from participants, we provided an text box in the questionnaire, allowing them to self-identify their gender freely rather than selecting from predefined categories.

During the experiment, participants received training on how to use the system and solve mathematical problems with MathBoard. Researchers recorded detailed interaction data with the system, including the number of dialogue rounds and problem-solving efficiency. This data helps to deeply understand the practical effects of the teaching aid. The results showed that 90% of the participants came from a teacher-type professional background, and 30% had a professional background related to mathematics. Ultimately, 34 participants completed the entire experimental process and provided effective data. We developed MathBoard based on ChatGPT-4o, which is provided by OpenAI under its terms of service, and its use is governed by those terms. The experiments were conducted on a device with an AMD Ryzen 9 7945HX processor and 16GB of RAM.

## C   Limitation

Although this study provides valuable insights into multimodal learning in elementary mathematics education and demonstrates the effectiveness of the MathBoard system in reducing cognitive load and promoting social construction, several limitations should be acknowledged and addressed in future research. The following section outlines these limitations.

First, the study is limited by a relatively small sample size. The findings are based on a sample of 34 pre-service teachers, which may restrict the generalizability of the results. Future studies should consider using a larger and more diverse sample to gain more comprehensive insights into the effectiveness of the proposed system. Furthermore, the current system is designed specifically for elementary mathematics, which limits its scalability to higher education and other subjects. Future research should explore how this system can be adapted and applied to broader educational contexts. For instance, developing different board-writing generation methods for various subjects or use cases could significantly enhance its scalability.

In this study, we observed that multimodal learning supported by LLMs can enhance learners' social construction, contributing to improved learning outcomes. However, the long-term effects and mechanisms of LLM-supported multimodal learning on learners' development remain unclear and warrant further investigation. For example, while multimodal information reduces cognitive load for learners, it may enhance metacognitive activities and improve learning outcomes for some. Conversely, other learners may experience good results when using the MathBoard system but struggle to perform independently once the system is removed, due to a sudden increase in cognitive load. This could lower test scores and foster dependency on the system. These hypotheses are intriguing and deserve further exploration.

MathBoard represents an innovative exploration of LLM-supported multimodal learning. In future research, we intend to extend the foundational framework of MathBoard to other grade levels, subjects, and educational fields to enhance its applicability in broader educational contexts. Moreover, the issues of data privacy and the ethical implications of using large language models in education are critical and require further discussion. Given the sensitivity of educational data, future studies should focus on ensuring privacy protection and addressing ethical considerations when employing such technologies in the classroom. We look forward to further innovations and the advancement of LLM-supported multimodal learning, bringing us closer to realizing the vision of large-scale, personalized education.

