# OpenReview forum: "Synchronizing Verbal Responses and Board Writing for Multimodal Math Instruction with LLMs"
_NeurIPS.cc/2024/Workshop/MATH-AI — MATH-AI 24_

### Official Review · Reviewer_wsUA · 2024-10-07
**A novel system designed to enhance elementary mathematics instruction through the synchronization of verbal responses and board writing (BW) using large language models (LLMs).**

**Rating:** 7
**Confidence:** 4

**Review:**

Through a well-designed empirical study involving pre-service teachers, the authors demonstrate that MathBoard significantly improves engagement and the social construction of knowledge compared to traditional verbal-only teaching methods. The work is well-articulated, addressing both theoretical foundations and practical implementations, though it is currently limited to elementary-level math instruction.
The technical quality of the paper is solid. The methodology is clearly explained, especially in the sections on cross-modal interaction. The work is original in that it combines LLMs with BW in a cross-modal system specifically designed for elementary-level math education. While similar concepts exist, the integration of real-time verbal and visual synchronization within the educational context is promising.
The manuscript is generally clear and well-structured.

Pros:
- Multimodality of the paper, enhances social construction of knowledge and learner engagement.
- Novel integration of verbal and visual teaching with GenAI
- Grounding in Schema Theory and alignment with Social Constructivism.

Cons:
- Limited Scope
- Focuses on immediate learning outcomes
- Small Sample size

Overall, despite some limitations, the promising results and strong theoretical underpinnings position MathBoard as a valuable tool for enhancing mathematics education.

---

### Official Review · Reviewer_7RrT · 2024-10-07
**This paper proposes a cross-modal interaction approach combining language responses generated by large language models (LLMs) with board writing (BW), which can better facilitate students' knowledge construction. Compared to solely verbal responses, this cross-modal interaction is considered more effective and engaging.**

**Rating:** 5
**Confidence:** 2

**Review:**

Pros:
This paper develops a mathematics instructional strategy grounded in Schema Theory, which enhances the system's scalability and adaptability, making it more suitable for widespread application in real teaching scenarios.


Cons:
The experimental setup and main results are a bit unclear to me. Could you clarify the main findings of your paper? If you can explain these clearly, I would be happy to improve my score.

---

### Official Review · Reviewer_cUvd · 2024-10-07
**A promising framework for integrating LLMs for math-instruction but lacking clarity in the demonstration of the study**

**Rating:** 6
**Confidence:** 3

**Review:**

### Clarity

#### Pros
- The paper has distinct sections explaining the methodology and results.
- The research questions are clearly stated, giving the reader a clear context about the study's goal and contribution.
- The use of diagrams to explain the workflow and user interface was helpful.

#### Cons
- Some of the sections were missing (related work, experiment design) or short / insufficiently explained.
- There was no section of related work, giving no context to the previous and existing work about using LLMs to augment learning and education.

### Quality

#### Pros
- The framework of the MathBoard follows a well-defined experiment using ETAS-M as a baseline to assess MathBoard's impact.
- The high Cronbach's alpha shows that the methodology and analysis were consistent.

#### Cons
- As stated by the authors of the study, the sample size of only 34 participants being limited, will limit the generalizability of the findings.
- Since this study focuses on short-term learning outcomes, the long-term impact on cross-modal learning is an open question.
- While the paper does adhere to ethical guidelines, there is limited discussion on data privacy and the ethical implications related to the use of LLMs in education, which could be a critical point of consideration for such studies.

### Originality

#### Pros
- The idea of synchronizing verbal responses with Board Writing (BW) via an LLM-driven system is novel, particularly for early education and instruction.

#### Cons
- Although MathBoard introduces synchronized BW, there are existing systems that are somewhat explored in the educational context. It would have been helpful for the authors to elaborate more on the differentiating aspects of MathBoard.

### Significance

#### Pros
- The study shows a clear impact on demonstrating the capabilities to enhance learning, memorization, and cognitive advantages of using cross-modal learning approaches in education.
- This study has strong potential to be expanded to other educational instruction settings and subjects, contributing to AI-driven educational designs.

#### Cons
- The focus on elementary math education limits the broader impact of the work. Would have been beneficial for authors to (briefly) address how the foundational framework can be explored further for other subjects/fields within education.
- Unclear how the system can be sustained in the long-term, particularly with evolving AI models and the growing educational needs, reducing its immediate significance in practical applications.

---

### Official Review · Reviewer_153D · 2024-10-08
**Review for Synchronizing Verbal Responses and Board Writing for Cross-Modal Math Teaching with LLMs**

**Rating:** 7
**Confidence:** 3

**Review:**

Quality:
An empirical study involving 34 pre-service teachers assesses MathBoard's effectiveness using the Educational Technology Acceptance & Satisfaction Model (ETAS-M). The study demonstrates that MathBoard significantly improves learner engagement and social constructivism compared to traditional text-only interactions. The sample size is relatively small and homogeneous, which may affect the generalizability of the findings. Additionally, the technical details of the multi-agent system and the synchronization algorithms are not thoroughly explained, limiting reproducibility and understanding of the system's inner workings.

Clarity:
The paper is generally well-structured and written clearly. The introduction provides a good background and clearly states the research questions:
- How can the generative capabilities of LLMs be leveraged to provide cross-modal guidance in mathematics learning?
- Is the proposed cross-modal teaching method more acceptable and engaging for learners?
- Does the integration of Board Writing in mathematics instruction foster learners’ social construction of knowledge?

Originality:
While prior studies have explored LLMs in education and collaborative environments, the cross-modal synchronization to mimic human teaching strategies seems like a novel approach.

Significance:
The work has potential to impact AI-assisted education by enhancing learning experiences through reduced cognitive load and improved engagement.

Pros:
- MathBoard synchronizes LLM-generated verbal responses with Board Writing, closely emulating human teaching practices.
- The study provides quantitative evidence using ETAS-M, showing significant improvements in learner engagement and social constructivism (e.g., p = 0.010 for the social constructivism dimension).
- Participants reported increased willingness to participate in group activities and discussions when using MathBoard.
- The system's effectiveness was consistent across gender and teaching experience, suggesting broad applicability.
- Detailed descriptions and visual aids help users understand and interact with the system effectively.

Cons
- Currently applicable only to elementary mathematics, limiting scalability to higher education or other subjects.
- Insufficient explanation of the multi-agent system's architecture and synchronization algorithms.

---

### Decision · Program_Chairs · 2024-10-09

Accept